# Gut Inflammation Induced by Finasteride Withdrawal: Therapeutic Effect of Allopregnanolone in Adult Male Rats

**DOI:** 10.3390/biom12111567

**Published:** 2022-10-26

**Authors:** Silvia Diviccaro, Silvia Giatti, Lucia Cioffi, Eva Falvo, Monika Herian, Donatella Caruso, Roberto Cosimo Melcangi

**Affiliations:** Dipartimento di Scienze Farmacologiche e Biomolecolari, Università degli Studi di Milano, Via Balzaretti 9, 20133 Milan, Italy

**Keywords:** pro-inflammatory cytokines, serotonin, dopamine, gut steroids, pregnenolone, GABA-A receptor, post-finasteride syndrome

## Abstract

The treatment with finasteride (i.e., an inhibitor of 5α-reductase) may be associated with different side effects (i.e., depression, anxiety, cognitive impairment and sexual dysfunction) inducing the so-called post finasteride syndrome (PFS). Moreover, previous observations in PFS patients and an experimental model showed alterations in gut microbiota populations, suggesting an inflammatory environment. To confirm this hypothesis, we have explored the effect of chronic treatment with finasteride (i.e., for 20 days) and its withdrawal (i.e., for 1 month) on the levels of steroids, neurotransmitters, pro-inflammatory cytokines and gut permeability markers in the colon of adult male rat. The obtained data demonstrate that the levels of allopregnanolone (ALLO) decreased after finasteride treatment and after its withdrawal. Following the drug suspension, the decrease in ALLO levels correlates with an increase in IL-1β and TNF-α, serotonin and a decrease in dopamine. Importantly, ALLO treatment is able to counteract some of these alterations. The relation between ALLO and GABA-A receptors and/or pregnenolone (ALLO precursor) could be crucial in their mode of action. These observations provide an important background to explore further the protective effect of ALLO in the PFS experimental model and the possibility of its translation into clinical therapy.

## 1. Introduction

Finasteride is an inhibitor of the enzyme 5α-reductase clinically used for benign prostatic hyperplasia and androgenetic alopecia [1,2]. Despite its efficacy in these disorders, the drug treatment is also associated with different side effects in the sexual and psychological domains [3,4]. Importantly, recent observations have demonstrated that, in the case of androgenetic alopecia, side effects associated with finasteride may also persist after the drug suspension, inducing the so-called post-finasteride syndrome (PFS) [3,4,5]. In particular, PFS patients report psychiatric and andrological dysfunctions, associated with alterations in neuroactive steroid levels both in plasma and in cerebrospinal fluid (CSF) compared with healthy age-matched controls [6]. Additionally, observations obtained in an experimental model of PFS showed that finasteride has broad persistent consequences not only in the plasma and CSF, but also in the brain [7]. Interestingly, these persistent alterations in the central nervous system were associated with neuroinflammation, gliosis, depressive-like behavior and a decrease in adult hippocampal neurogenesis [8]. Unsurprisingly, due to the existence of the gut-brain axis [9,10], finasteride treatment and its withdrawal can also affect the gut microbiota composition both in the experimental model [8] and in PFS patients [11]. For instance, immediately after drug treatment, an increase in the *Bacteroidetes* phylum as well as in the *Prevotellaceae* family was reported in the PFS experimental model [8], while the discontinuation of the finasteride treatment induced a decrease in the *Ruminococcaceae* family, *Oscillospira* and *Lachnospira* genus [8]. Changes in gut microbiota could be ascribed to different factors, such as changes induced by finasteride in plasma and/or brain neuroactive steroid levels [7]. Indeed, gonadectomy and hormone replacement have a clear effect on the gut bacteria in rodents [12,13,14,15,16,17,18,19]. It is also important to note that the gut itself is able to synthesize steroid molecules [20] and, as recently reported, the adult male rat colon expresses steroidogenic enzymes, also including 5α-reductase. Consequently, it is able to synthesize and metabolize steroids such as progesterone (PROG), testosterone (T) and 17beta-estradiol (17β-E) [20]. However, whether finasteride treatment may affect steroid levels in the male colon is still unknown. Additionally, the change in the gut microbiota composition previously observed [8] seems to suggest an inflammatory microenvironment with putative consequences on gut physiology [21,22]. In this context, it is also important to highlight that serotonin, which plays a crucial role in the regulation of several physiological functions, such as motility, secretion and visceral sensitivity [23], is also related to inflammatory responses in the gut [24,25]. Moreover, gut microbiota also has the potential to influence the levels of other neurotransmitters, for instance dopamine [26], which is widely distributed in the intestinal tract and affects gastric secretion and motility [27,28]. Furthermore, the altered microbial composition, termed dysbiosis, has been implicated in mucosal barrier dysfunction and inflammatory cytokine production [29], which are partners in leaky gut syndrome affecting the gut homeostasis [30].

For all these reasons, we have explored the effect of chronic treatment with finasteride (i.e., for 20 days) and its withdrawal (i.e., for 1 month) on the levels of steroids and neurotransmitters, pro-inflammatory cytokines and gut permeability markers in the adult male colon. In particular, the levels of steroids, such as pregnenolone (PREG), PROG, dihydroprogesterone (DHP), allopregnanolone (ALLO), isoallopregnanolone (ISOALLO), dehydroepiandrosterone (DHEA), T, dihydrotestosterone (DHT), 5α-androstane-3α, 17β-diol (3α-diol) and 17β-E, and of neurotransmitters, such as dopamine and serotonin, have been assessed by liquid chromatography-tandem mass spectrometry. The gene expression of tool-like receptor 4 (TLR-4), interleukin 1 beta (IL-1β), tumor necrosis factor alpha (TNF-α), interleukin 6 (IL-6), zonulin 1 (ZO-1) and claudin 1 (Cld-1) were also evaluated. Based on the results obtained in this first analysis, we further reported that ALLO treatment is able to recover from gut inflammation observed at the finasteride withdrawal in adult male rats.

## 2. Materials and Methods

### 2.1. Animals

Sprague Dawley male rats (200−225 g at arrival, Charles River Laboratories, Lecco, Italy) were used in these experiments. All procedures were performed in accordance with national (D.L. No. 26, 4 March 2014, G.U. No. 61 14 March 2014) and international laws and policies (EEC Council Directive 2010/63, 22 September 2010: Guide for the Care and Use of Laboratory Animals, United States National Research Council, 2011) and were previously approved by the local ethics committee, and by the Italian Ministry of Health (authorization 261-2021-PR). The animals were housed in the animal care facility of the Dipartimento di Scienze Farmacologiche e Biomolecolari (DiSFeB) at the Università degli Studi di Milano, Italy. The rats were acclimated to the new environment for one week. At sacrifice, the animals were placed individually in an induction chamber, and anesthesia was induced with 2% isoflurane (ISO VET, La Zootecnica, Pavia, Italy) until the loss of the righting reflex. Then, after sacrifice, the plasma and colon were stored immediately at −80 °C until the analysis.

### 2.2. Treatments

In the first experiment, male rats were treated with finasteride or vehicle (Control) and sacrificed at 24 h and 1 month after the last treatment. Finasteride (3 mg/kg/day; Sigma-Aldrich, Milan, Italy) was suspended in a vehicle solution of sesame oil and ethanol (5% *v*/*v*). Either this solution or vehicle was administered to the animals subcutaneously, at a volume of 100 μL/day for 20 days.

In the second experiment, male rats were treated with ALLO or vehicle (Control) during the withdrawal period starting 14 days after the last finasteride injection. The animals were sacrificed 24 h after the last ALLO treatment. ALLO (1 mg/rat/day; Sigma-Aldrich, Milan, Italy) was suspended in a vehicle solution of sesame oil and ethanol (5% *v*/*v*). Either this solution or vehicle was administered to the animals subcutaneously, at a volume of 100 μL/day every other day for sixteen days (eight treatments).

### 2.3. Steroid Level Evaluation by Liquid Chromatography Tandem Mass Spectrometry Analysis

For the quantitative analysis of steroids, longitudinal sections of the colon (200 mg/sample) and plasma (300 μL/sample) were collected and internal standards, 17β-E-2,3,4-^13^C_3_ (2 ng/sample), PROG-2,3,4,20,25-^13^C_5_ (0.4 ng/sample) and PREG-20,21-^13^C_2_-16,16 D_2_ (10 ng/sample) were added. Tissue samples were homogenized using the Tissue Lyser (Qiagen, Italy), in ice-cold methanol/acetic acid 1% and purified by the organic phase extraction as previously described [31,32]. Plasma samples were extracted and purified by organic phase extraction as previously described [31,32,33]. The quantitative analysis was performed using a linear ion trap-mass spectrometer LTQ (Thermo Fisher Scientific, Waltham, MA, USA), equipped with a Surveyor Liquid Chromatography (LC) Pump Plus and a Surveyor Autosampler Plus (Thermo Fisher Scientific, Waltham, MA, USA), operating in positive atmospheric pressure chemical ionization (APCI+) mode. The chromatographic separation was achieved with a Hypersil Gold column C18 (100 × 2.1 mm, 3 μm; Thermo Fisher Scientific, Waltham, MA, USA) maintained at 40 °C, equipped with Hypersil Gold DROP-IN GUARD (10 × 2.1 mm, 3 μm Thermo Fisher Scientific, Waltham, MA, USA). The mobile phase consisted of 0.1% formic acid in water and 0.1% formic acid in methanol in a gradient elution at a flow rate of 0.300 mL/min. LC-MS/MS data were evaluated using Excalibur^®^ release 2.0 SR2 (Thermo Fisher Scientific, Waltham, MA, USA). Quantitative analysis of PREG, PROG, DHP, ALLO, ISOALLO, DHEA, T, DHT, 3α-diol and 17β-E was achieved on the basis of calibration curves freshly prepared.

### 2.4. Dopamine and Serotonin Level Evaluation

For the quantitative analysis of catecholamine (i.e., dopamine and serotonin), 200 mg of colon samples were analyzed as previously reported [34]. In brief, the tissue was homogenized using the Tissue Lyser II (Qiagen, Hilden, Germany) in ice-cold methanol supplemented with an internal standard, dopamine-1,1,2,2-d_4_ hydrochloride (500 ng/sample). Then, to remove particulate matter, samples were centrifuged at 14,000 rpm for 20 min at 4 °C and the supernatant was evaporated to dryness under a stream of nitrogen. The dry pellet was reconstituted with 300 μL of water, vortex-mixed for 10 s, and 300 μL of chloroform−isopropanol (70:30, *v*/*v*) was added. After mixing, the upper aqueous layer was filtered with a 0.2 μm filter (SRC grade, regenerated cellulose membrane filter, CHMLAB Group). For the quantitative analysis, 5 μL/sample was injected in API 3500 (AB Sciex, Framingham, MA, USA) mass spectrometer, equipped with an electrospray source (ESI+) and triple quadrupole analyzer, interfaced with a pump for the HPLC model EXION SL (AB Sciex, Framingham, MA, USA). To obtain the chromatographic separation of the analytes, a column for HPLC Luna Omega 5 μm PS C18 100 Å was used (Phenomenex, Torrance, CA, USA). Quantitative analysis of dopamine and serotonin was achieved based on the calibration curves freshly prepared.

### 2.5. Real-Time Polymerase Chain Reaction

Total RNA from tissues was extracted using the standard Trizol protocol, in accordance with the manufacturer’s protocol (Invitrogen, San Giuliano Milanese, Italy) and prepared using the Direct-zol TM RNA MiniPrep kit (Zymo Research, Irvine, CA, USA). After quantification, RNA was analyzed using a TaqMan quantitative real-time PCR instrument (CFX96 Real Time system; Bio-Rad Laboratories, Segrate, Italy) using the Luna Universal One-Step RT-qPCR Kit (New England BioLabs Inc., Ipswich, MA, USA). The samples were run in 96-well formats in duplicate as multiplexed reactions with a normalizing internal control, 36B4. Specific TaqMan MGB probe and primer sequences were purchased at Eurofins MWG-Operon (Milano, Italy) and are available on request:

36B4 (Z29530.1) fwd: GGATGACTACCCAAAATGCTTC; rev: TGGTGTTCTTGCCCATCAG; TLR-4 (NM_019178.1) fwd: CATGACATCCCTTATTCAACCAAG; rev: GCCATGCCTTGTCTTCAATTG; IL-1β (NM_031512.2) fwd: TGCAGGCTTCGAGATGAAC; rev: GGGATTTTGTCGTTGCTTGTC; TNF-α (NM_012675.3) fwd: CTTCTCATTCCTGCTCGTGG; rev: TGATCTGAGTGTGAGGGTCTG; IL-6 (NM_012589.1) fwd: AAGCCAGAGTCATTCAGAGC; rev: GTCCTTAGCCACTCCTTCTG; subunit α1 (NM_183326.2) fwd: GAGAGTCAGTACCAGCAAGAAC; rev: AGAACACGAAGGCATAGCAC; subunit α3 (NM_017069.3) fwd: TTCACTAGAATCTTGGATCGGC; rev: TCTGACACAGGGCCAAAAC; subunit β2 (NM_012957.2) fwd: CTGGATGAACAAAACTGCACG; rev: ACAATGGAGAACTGAGGAAGC of GABA-A receptor. LIFE TECHNOLOGIES ITALIA (Milano, Italy): subunit α4 (Rn00589846_m1), subunit β3 (Rn00567029_m1), subunit δ (Rn01517017_g1), subunit γ2 (Rn00788325_m1) of GABA-A receptor; Zonulin-1 (ZO-1, Rn02116071_s1) and Claudin-1 (Cldn-1, Rn00581740_m1).

### 2.6. Statistics

LC-MS/MS and real-time PCR analysis were analyzed by an unpaired two-tailed Student’s *t*-test, after checking for normal distribution with the Kolmogorov–Smirnov test. *p* < 0.05 was considered significant. The effect of the treatment of ALLO was analyzed using one-way ANOVA followed by Uncorrected Fisher’s LSD post hoc test. Analyses were performed using Prism, version 7.0a (GraphPad Software Inc., San Diego, CA, USA). Linear regression analysis and Pearson’s correlation coefficient were computed to assess the potential relationship between 2 different variables.

## 3. Results

In the first experiments, the levels of steroids were evaluated in adult male rat colons after a finasteride long-term treatment (Figure 1) and at withdrawal (Figure 2). As reported in Figure 1, treatment for 20 days with finasteride was able to significantly decrease the ALLO levels. Finasteride subchronic treatment also affected androgens. As expected, an increase of T associated with a decrease in its metabolites, such as DHT and 3α-diol, were reported. The levels of PREG, PROG, DHP, DHEA and 17β-E were not significantly modified (Figure 1).

The assessment of these steroids after 1 month of withdrawal revealed that ALLO levels were still significantly decreased, whereas an increase in PREG levels was observed (Figure 2).

The levels of the other steroids measured at the finasteride withdrawal were not significantly modified. Data reported in Figure 3 indicate that finasteride treatment did not affect the gene expression of pro-inflammatory cytokines (panel A), or gut permeability markers (panel C), as well as the levels of dopamine and serotonin (panel E) in adult male rat colons. In contrast, finasteride withdrawal induced a significant increase in the mRNA levels of IL-1β and TNF-α, with no changes in TLR-4 and IL-6 levels (panel B) and in those of ZO-1 and Cld-1 (panel D). The levels of dopamine and serotonin significantly decreased and increased, respectively (panel F).

It is interesting to note that PREG levels at the finasteride withdrawal positively correlated with mRNA levels of IL-1β (*p* = 0.002, Pearson’s r = +0.82), TNF-α (*p* = 0.013, Pearson’s r = +0.82) and negatively with dopamine (*p* = 0.004, Pearson’s r = −0.88) but not with serotonin levels (*p* = 0.186).

Based on the reported anti-inflammatory features of ALLO [35,36,37,38,39], using a previously established treatment schedule for steroids [40,41,42,43], we have analyzed the possible protective effects of this steroid on changes induced by finasteride withdrawal. As reported in Figure 4, ALLO treatment was able to significantly counteract the increase induced by finasteride treatment in mRNA levels of IL-1β (panel A) and TNF-α (panel B), as well as in the levels of serotonin in the adult male rat colons (panel D). ALLO treatment did not counteract the decrease in the dopamine levels induced by finasteride withdrawal (panel C).

ALLO is a potent ligand of the GABA-A receptor [38,44,45]. Therefore, the gene expression of some subunits of the GABA-A receptors was evaluated in adult male rat colons. As reported in Figure 5, finasteride withdrawal significantly decreased the mRNA levels of subunits α3, β2, and β3, whereas an increase in the δ subunit occurred. The ALLO treatment was able to significantly counteract changes in the β2, β3 and δ subunits.

However, the ALLO effects might also be mediated by other steroids. Therefore, we assessed at the finasteride withdrawal the metabolic fate of ALLO treatment by LC-MS/MS in adult male rat colons. As reported in Figure 6, an expected significant increase in the ALLO levels occurred after the steroid treatment.

Interestingly, the observed increase in PREG levels induced by finasteride withdrawal was significantly counteracted by the ALLO treatment. This effect is coupled with changes in the gene expression of enzyme converting cholesterol into PREG (i.e., P450 side-chain cleavage, P450scc). Indeed, the mRNA levels of this enzyme were significantly increased by finasteride withdrawal and significantly decreased by ALLO treatment (Control: 0.749 ± 0.099 vs. Finasteride: 1.227 ± 0.323, n = 7 for each group, *p* = 0.0028; ALLO: 0.689 ± 0.285, n = 7 for each group, *p* = 0.006 vs. finasteride-treated rats).

## 4. Discussion

The data reported indicate that subchronic treatment with finasteride affects the levels of some steroids in the colon of adult male rats. Indeed, as reported, a significant decrease in ALLO levels and an increase in T levels associated with a decrease in its 5α-reduced metabolites (i.e., DHT and 3α-diol) occurred after finasteride subchronic treatment for 20 days (Figure 1). Interestingly, as previously reported, subchronic treatment with finasteride also decreased the 3α-diol levels in the cerebellum and the CSF, as well as the levels of DHT in plasma [7]. However, after finasteride treatment, other steroids are affected in the brain [7] and not in the gut. At the finasteride withdrawal, the steroid levels in adult male rat colons showed both common and peculiar effects compared with that observed after the drug treatment. Indeed, the ALLO levels still decreased, while PREG levels significantly increased (Figure 2). Because PREG is the first steroid synthesized by cholesterol, and consequently substrate of all steroids, including ALLO, the increase in PREG observed at the finasteride withdrawal might be interpreted as an attempt to counteract the decrease in ALLO levels.

Following finasteride treatment, steroid changes occurring at drug withdrawal in the colon did not exactly reflect the changes occurring in brain areas [7], further confirming that steroid levels in the gut-brain axis are differently affected by finasteride depending on the structure considered. Indeed, ALLO levels decreased in the cerebral cortex, but not in the cerebellum and hippocampus, while PREG levels increased in the cerebellum, decreased in the hippocampus and were unmodified in the cerebral cortex [7]. To explore the consequences of local changes in steroid levels in adult male rat colons, we evaluated in this tissue the gene expression of TLR-4 and pro-inflammatory cytokines, such as IL-1β, TNF-α and IL-6, of markers of gut permeability, such as ZO-1 and Cldn-1, and levels of neurotransmitters, such as dopamine and serotonin. As reported, subchronic treatment with finasteride did not affect these markers, but an increase in IL-1β and TNF-α as well as a decrease in dopamine levels and an increase in those of serotonin were reported at the drug withdrawal in the colon of adult male rats. These changes, as reported by others [46,47,48,49] may suggest a local inflammation. Indeed, in patients with irritable bowel syndrome (IBS), there is a decreased transcription of the serotonin transporter (SERT) resulting in elevated serotonin level, which ultimately causes diarrhea and discomfort, which is transmitted by serotonin through the gut-brain axis [50,51]. Gut inflammation was also supported by our previous observations in this PFS experimental model, indicating alterations in gut microbiota populations at the finasteride withdrawal, with specific significant changes in the microbial communities (weighted and unweighted UniFrac distance) [8]. In particular, we reported that, after therapy discontinuation, the phylum of *Firmicutes* decreased, whereas *Bacteroidetes* increased compared with basal values in rats [8]. *Bacteroidetes* bacteria are often increased in inflammatory bowel disease (IBD) and associated with its progression and development [52]. For instance, mucosal biopsies from inflamed and non-inflamed regions of the intestine from patients with IBD and healthy individuals revealed increased *Bacteroidetes* and reduced Firmicutes abundance [53]. Certain gut bacterial species can adhere to the gut mucosa and invade mucosal epithelial cells, which results in an inflammatory response mediated by the production of TNF-α by monocytes and macrophages [54].

Moreover, increased levels of L-dopa and decreased levels of dopamine were reported in patients with IBD, indicating low L-amino acid decarboxylase activity [55]. Impairment of the dopaminergic system as a feature of IBD pathogenesis is supported by the finding that dopamine agonists may rescue to the normal function [56].

Furthermore, a decrease in the *Ruminococcaceae* family, *Oscillospira* and *Lachnospira* genus, species that are related to inflammatory processes [21,22,37,57,58,59], was also observed at the finasteride withdrawal [8].

We also demonstrated that, at the finasteride withdrawal, ALLO treatment is protective of the alterations occurring in adult male rat colons. Indeed, this steroid, as reported in other experimental models expresses anti-inflammatory features [35,36,37,38,39], counteracts the increase in IL-1β and TNF-α gene expression, as well as that in the levels of serotonin in the adult male rat colon. Based on the protective effects exerted by ALLO treatment in gut inflammation induced by finasteride withdrawal, we first evaluated whether this effect was related to the capacity of this steroid to modulate GABA-A receptors. The GABA-A receptor has a pentameric structure formed by multiple subunits. Nineteen subunits have been identified and among these, α, β, δ and γ subunits are more widely expressed and are targets for steroids, such as the ALLO [44,60]. GABA-A receptors are enriched within the enteric nervous system [61] regulating stress-induced gastrointestinal inflammation [62,63]. The data obtained indicate that changes in the gene expression of some subunits of GABA-A receptor in adult male rat colons occurred at the finasteride withdrawal (i.e., α3, β2, β3 and δ). Interestingly, a decrease in the β3 subunit, associated with a decrease in ALLO levels, also occurred in the cerebral cortex at the finasteride withdrawal [7]. ALLO treatment counteracted changes in the β2, β3 and δ subunits present in adult male rat colons, suggesting that the GABA-A receptor plays a role in the protective effect of ALLO reported.

In addition, ALLO treatment may also influence the levels of other steroids present in the gut. Indeed, as reported, the steroid treatment was able to significantly counteract the increase in PREG levels occurring at the finasteride withdrawal. Interestingly, the decrease in the PREG levels induced by ALLO treatment was associated with a decrease in the gene expression of the enzyme converting cholesterol into PREG (i.e., P450scc), suggesting a specific tissue regulation in the synthesis of this steroid. This concept is further supported by the finding that PREG levels in the plasma of male rats at the finasteride withdrawal were not affected by ALLO treatment (data not shown). To our knowledge, this is the first demonstration that ALLO treatment is able to control PREG synthesis in the gut. As we demonstrated, the increase in the PREG levels at the withdrawal was positively correlated with inflammation. Indeed, due to the anti-inflammatory features of this steroid [64,65], it is possible to hypothesize that the observed PREG increase could be ascribed to a possible compensatory anti-inflammatory response, to cope with the negative pattern also induced by finasteride withdrawal. Interestingly, a very similar increase in PREG levels in the rat gut has also been reported at paroxetine withdrawal [66]. Indeed, similarly to what we observed with the finasteride, the suspension of this anti-depressive drug induced an inflammatory environment in the colon of adult male rats [66].

Altogether, these results indicate a local relationship between PREG and ALLO in the rat colon. Indeed, PREG, as a substrate of ALLO, increased at the finasteride withdrawal to cope with the decrease of ALLO, and *vice versa*, ALLO treatment increased the low levels of the steroid in the gut, inhibiting PREG levels.

## 5. Conclusions

Data obtained indicate that finasteride withdrawal induces gut inflammation in adult male rats and that ALLO treatment is able to counteract some of these alterations, such as the increase in the levels of pro-inflammatory cytokines (i.e., IL-1β and TNF-α) and serotonin. As mentioned above, patients treated with finasteride may show the so-called PFS, which is characterized by sexual side effects (i.e., low libido, erectile dysfunction, decreased arousal and difficulty in achieving orgasm), depression, anxiety and cognitive complaints. Importantly, finasteride withdrawal can also affect the gut microbiota composition in PFS patients [11] and its experimental model [8]. Therefore, because (i) sexual dysfunction may be related to alterations in gut microbiota [67,68,69,70] and (ii) the existence of the well-described gut-brain axis [9,10], observations here obtained may provide an important background to explore, with this experimental model, the protective effect of ALLO on psychiatric and andrological dysfunctions, laying the groundwork for possible therapy in PFS patients.

## Figures and Tables

**Figure 1 biomolecules-12-01567-f001:**
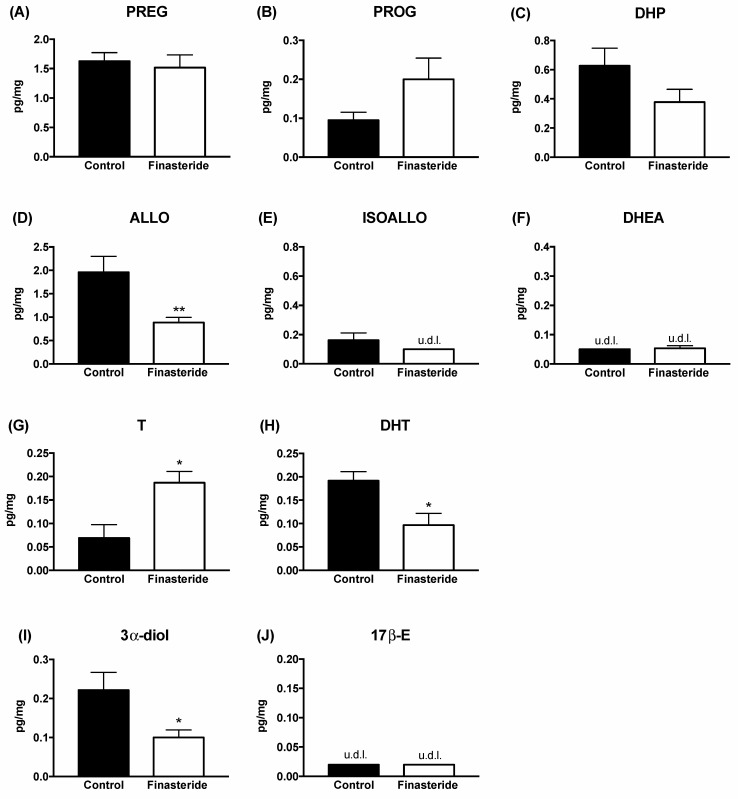
Levels of gut steroids in the colon of the control (black bar) and finasteride-treated (white bar) rats: effect after treatment. Pregnenolone (PREG, (**A**)), progesterone (PROG, (**B**)), dihydroprogesterone (DHP, (**C**)), allopregnanolone (ALLO, (**D**)), isoallopregnanolone (ISOALLO, (**E**)), dehydroepiandrosterone (DHEA, (**F**)), testosterone (T, (**G**)), dihydrotestosterone (DHT, (**H**)), 5α-androstane-3α, 17β-diol (3α-diol, (**I**)) and 17beta-estradiol (17β-E, **J**). Data are expressed as pg/mg ± SEM, n = 6 for each group. Unpaired Student’s *t*-test analysis: * *p* < 0.05, ** *p* < 0.01 vs. control rat colon. U.d.l. = under the detection limit. Detection limits were 0.05 pg/mg for dehydroepiandrosterone, 0.1 pg/mg for isoallopregnanolone and 0.02 pg/mg for 17β-Estradiol.

**Figure 2 biomolecules-12-01567-f002:**
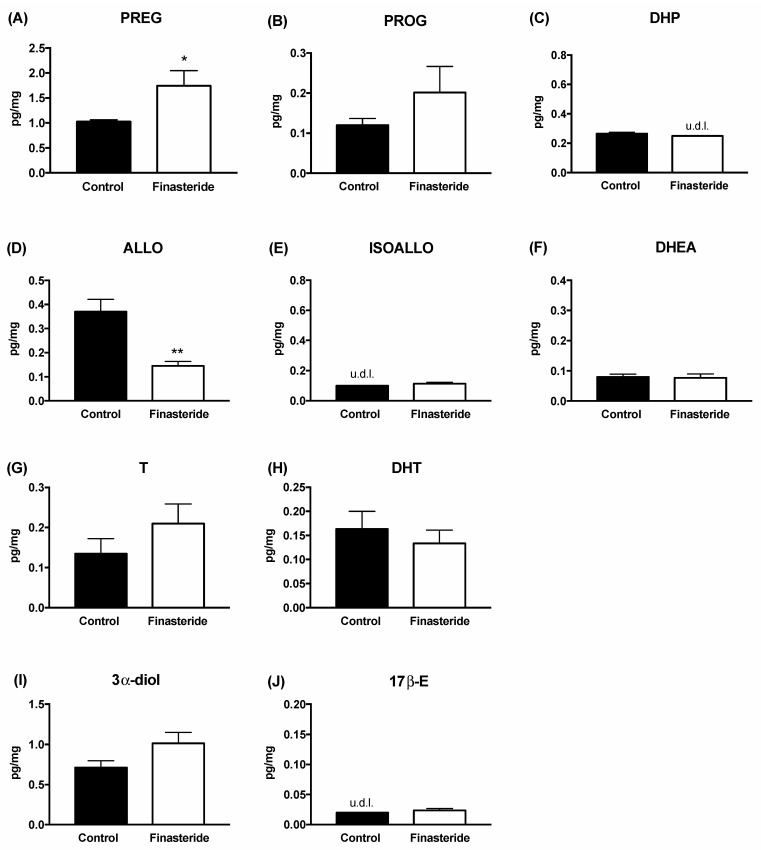
Levels of gut steroids in the colon of the control (black bar) and finasteride-treated (white bar) rats: effect at withdrawal. Pregnenolone (PREG, (**A**)), progesterone (PROG, (**B**)), dihydroprogesterone (DHP, (**C**)), allopregnanolone (ALLO, (**D**)), isoallopregnanolone (ISOALLO, (**E**)), dehydroepiandrosterone (DHEA, (**F**)), testosterone (T, (**G**)), dihydrotestosterone (DHT, (**H**)), 5α-androstane-3α, 17β-diol (3α-diol, (**I**)) and 17beta-estradiol (17β-E, (**J**)). Data are expressed as pg/mg ± SEM, n = 6 for each group. Unpaired Student’s *t*-test analysis: * *p* < 0.05, ** *p* < 0.01 vs. control rat colon. u.d.l. = under the detection limit. Detection limits were 0.05 pg/mg for dehydroepiandrosterone, 0.25 pg/mg for dihydroprogesterone, 0.1 pg/mg for isoallopregnanolone and 0.02 pg/mg for 17β-Estradiol.

**Figure 3 biomolecules-12-01567-f003:**
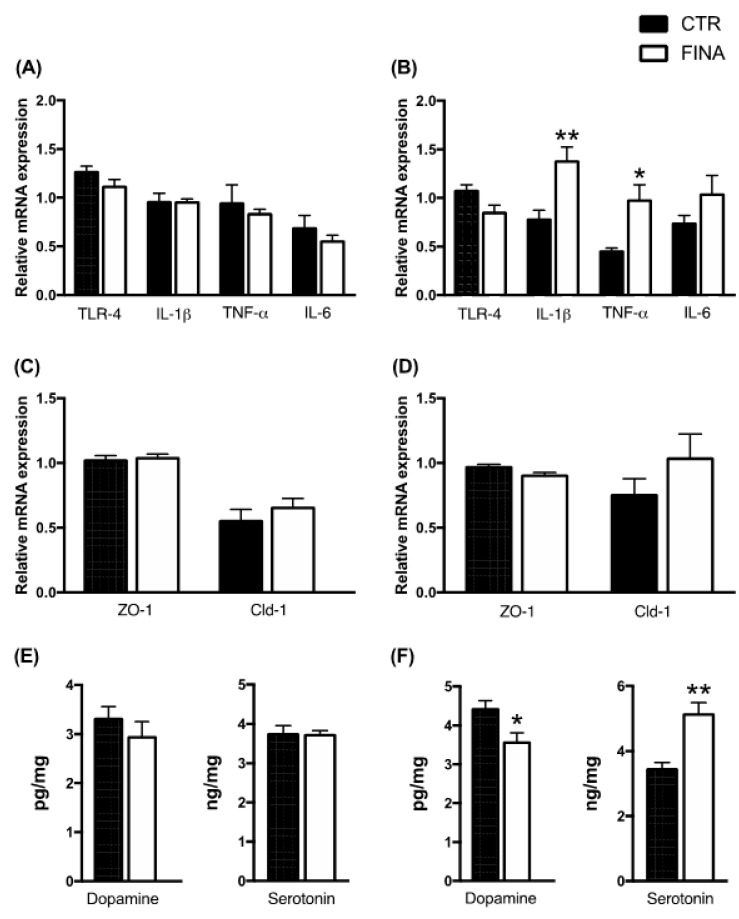
Gene expression of pro-inflammatory markers (**A**,**B**); gut permeability markers (**C**,**D**) detected by real-time PCR in colon of controls (CTR) and finasteride-treated rats (FINA) after drug treatment (**A**,**C**) and at withdrawal (**B**,**D**). The columns represent the mean ± SEM after normalization with 36B4 mRNA, n = 6 for each group. Levels of dopamine and serotonin (**E**,**F**) detected by LC-MS/MS in the colon of controls and finasteride-treated rats after drug treatment (**E**) and at withdrawal (**F**). Data are expressed in pg/mg or ng/mg ± SEM, n = 6 for each group. The unpaired Student’s *t*-test was used for statistical * *p* < 0.05, ** *p* < 0.01 vs. control group.

**Figure 4 biomolecules-12-01567-f004:**
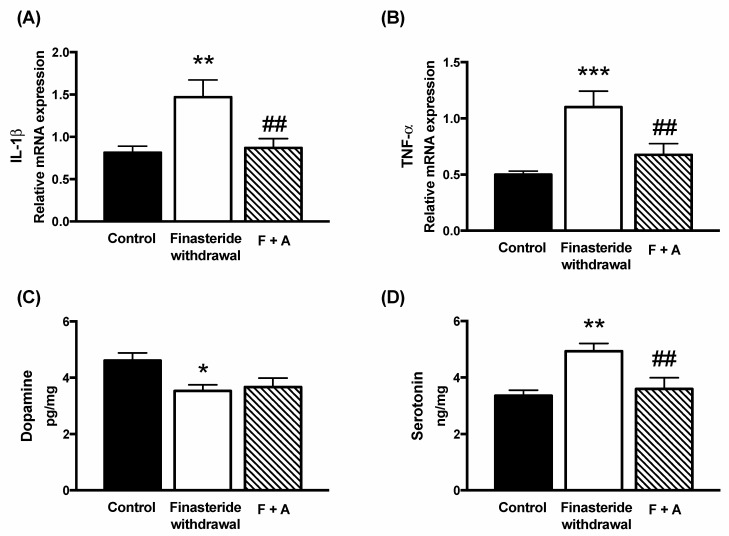
Gene expression of IL-1β (**A**) and TNF-α (**B**) detected by real-time PCR in the colon of controls, finasteride-treated rats and ALLO-treated rats after drug withdrawal (F + A). The columns represent the mean ± SEM after normalization with 36B4 mRNA. Levels of dopamine (**C**) and serotonin (**D**) were detected by LC-MS/MS in the colon of controls, finasteride-treated rats and ALLO-treated rats after drug withdrawal. Data are expressed in pg/mg or ng/mg ± SEM, n = 7 for each group. The effect of treatment of ALLO was analyzed using one-way ANOVA followed by Uncorrected Fisher’s LSD post hoc test (significance: * *p* < 0.05; ** *p* < 0.01; *** *p* < 0.001 vs. control group; ## *p* < 0.01 vs. finasteride-treated rats).

**Figure 5 biomolecules-12-01567-f005:**
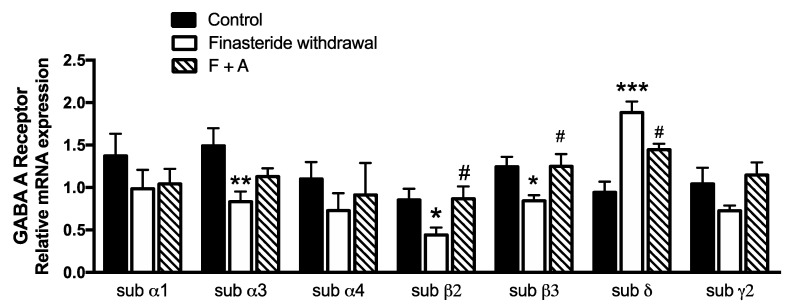
Gene expression of GABA-A receptor subunits α1, α3, α4, β2, β3, δ and γ2 in the colon of controls, finasteride-treated rats and ALLO-treated rats after drug withdrawal. The columns represent the mean ± SEM after normalization with 36B4 mRNA, n = 7 for each group. The effect of treatment of ALLO was analyzed using one-way ANOVA followed by Uncorrected Fisher’s LSD post hoc test (significance: * *p* < 0.05, ** *p* < 0.01, *** *p* < 0.001 vs. control group; # *p* < 0.05 vs. finasteride-treated rats).

**Figure 6 biomolecules-12-01567-f006:**
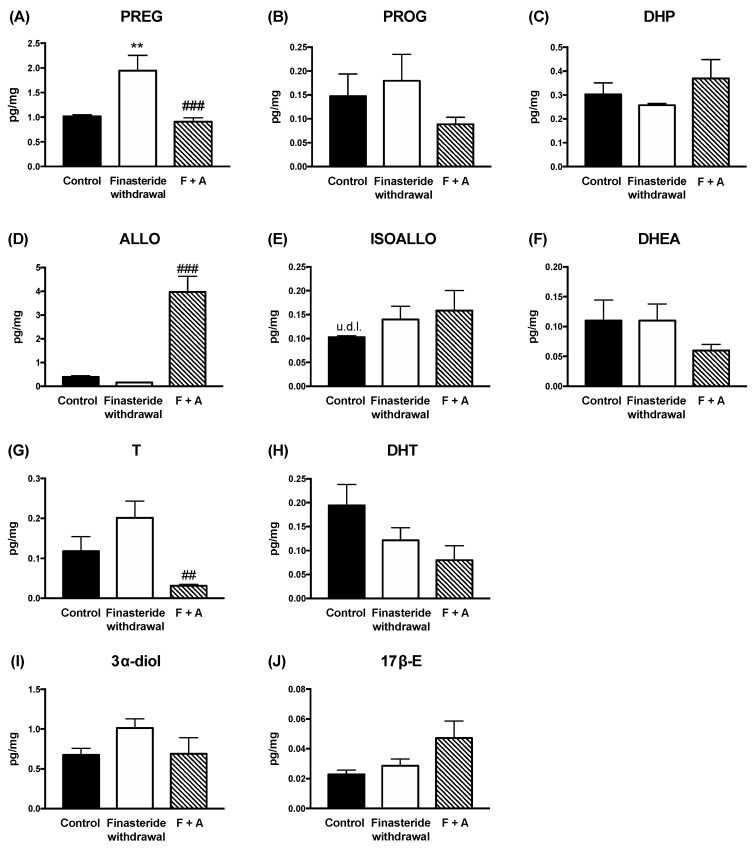
Levels of gut steroids in the colon of control, finasteride-treated rats and ALLO-treated rats: effect at withdrawal (F + A). Pregnenolone (PREG, (**A**)), progesterone (PROG, (**B**)), dihydroprogesterone (DHP, (**C**)), allopregnanolone (ALLO, (**D**)), isoallopregnanolone (ISOALLO, (**E**)), dehydroepiandrosterone (DHEA, (**F**)), testosterone (T, (**G**)), dihydrotestosterone (DHT, (**H**)), 5α-androstane-3α, 17β-diol (3α-diol, (**I**)) and 17beta-estradiol (17β-E, (**J**)). Data are expressed as pg/mg ± SEM, n = 7 for each group. The effect of treatment with ALLO was analyzed using one-way ANOVA followed by Uncorrected Fisher’s LSD post hoc test (significance: ** *p* < 0.01 vs. control group; ## *p* < 0.01; ### *p* < 0.001 vs. finasteride-treated rats). u.d.l. = under the detection limit. The detection limit was 0.1 pg/mg for isoallopregnanolone.

## Data Availability

Data are available on request.

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
