# Peer review of "Gut Inflammation Induced by Finasteride Withdrawal: Therapeutic Effect of Allopregnanolone in Adult Male Rats"

_biomolecules, 2022, doi:10.3390/biom12111567_

Round 1

Reviewer 1 Report

This is a well-designed study with strong rationale. The methods are appropriate and well-described. The results are interesting and provide novel and important information. The discussion is well-balanced.

My only comment has to do with lines 340-341, where the authors talk about a correlation between PREG and inflammation. I did not see a correlation reported in the results - this statistic should be added and then the discussion will accurately reflect the results.

Author Response

We thank the reviewer for the appreciation. The correlation of pregnenolone with inflammation was reported in the text under the result section (see lines 266-269). We did that in order to reduce at minimum the number of figures.

Reviewer 2 Report

I read your manuscript (Gut inflammation induced by finasteride withdrawal: therapeutic effect of allopregnanolone in adult male rat) and I think that it is very interesting because the trated topic is current. However, I suggest some modifies before the publication.

The abstract needs to add some background.

The current language is not enough for publication in this journal and some sentences are difficult to understand. Therefore, I suggest carefully revising the language throughout the manuscript. 

Some of the grammar in the manuscript needs revision.

I suggest adding limitations to the discussion or conclusion. This is something that must be done.

The discussion needs to be more in-depth and provide adequate reference support.

too many keywords, please revise.

The background, significance, and purposes of this study need to be discussed in more detail.

1. Line 59: “gut microbiota has the potential to influence also the levels of other neurotransmitters” should be written as “gut microbiota also has the potential to influence the level of other neurotransmitters”;

2. Line 119: “0.1% formic acid”;

3. Line 174: “In the first experiments” should be written as “In the first experiment”;

4. Line 198: “the gene expression of” should be written as “the gene expressions of”;

5. Line 327: “also” should written in front of the “occurred”;

6. Line 346 and 347: “also” should written in front of the“induced”.

Author Response

We thank the reviewer for the appreciation. The grammar of the manuscript has been further revised and your suggestions added; we have also implemented the abstract and conclusions as suggested (please see the track changes). For the number of keywords we have followed the author instruction.